# Optimizing Employee Creativity in the Digital Era: Uncovering the Interactional Effects of Abilities, Motivations, and Opportunities

**DOI:** 10.3390/ijerph17031038

**Published:** 2020-02-06

**Authors:** Wenjing Cai, Svetlana Khapova, Bart Bossink, Evgenia Lysova, Jing Yuan

**Affiliations:** 1School of Public Affairs, University of Science and Technology of China, Hefei 230026, China; 2Department of Management and Organization, Vrije Universiteit Amsterdam, 1081HV Amsterdam, The Netherlandss.n.khapova@vu.nl (S.K.); e.lysova@vu.nl (E.L.); 3Department of Science, Business and Innovation, Vrije Universiteit Amsterdam, 1081HV Amsterdam, The Netherlands; b.a.g.bossink@vu.nl; 4School of Foreign Studies, Anhui Sanlian University, Hefei 230601, China

**Keywords:** creativity, interactions, digital era, HRM, AMO theory

## Abstract

An increasing digitalization in all aspects of life and work reshapes traditional assumptions about human creativity. Both scholars and practitioners raise many questions with regards to how to stimulate employee creativity in the digital work context. While there are many studies that examine predictors of employee creativity, little effort has been made thus far to synthesize these findings in way that would provide meaningful guidance to organizations and to provide bases for future research. With this paper we aim to contribute to filling this gap. We systematically review empirical studies on predictors of employee creativity published in the past 30 years and organize findings following an established human resources management framework: Ability–Motivation–Opportunity (AMO) theory. This organizing framework enables us to clearly depict how contextual factors (a) separately and (b) jointly influence individual employee creativity. Specifically, it enables us to depict two possible models—combination and multiplicative models—through which contextual factors interact with individual factors in predicting employee creativity. Through synthesizing evidence for each of the models, we demonstrate to scholars and practitioners what is known about the interactional effects of contextual and personal factors on employee creativity, and what still needs to be studied if we are to take the field of research on creativity in the digital era forward.

## 1. Introduction

The era of increasing digitization is challenging traditional assumptions about how organizations can structure themselves to succeed in the context of global markets [1,2]. Among the key challenges is the question of how organizations structure the work of their employees to obtain the most innovative outcomes [3]. While much research suggests that the future workforce should be equipped with digital skills and mindsets, less is written about another human skill and competency that is also vital to the success of future organizations in the digital era, namely, ‘creativity’ [4].

Defined as the means by which individuals produce novel and useful ideas, products, and processes [5], creativity is needed to conceive and generate new working methods that are made possible and are initiated by digital transitions [6]. While the Oxford Martin School suggests that 57% of jobs in Organisation for Economic Cooperation and Development (OECD) countries will disappear due to automation in the next 20 years (e.g., 69% in India and 77% in China), scholars also note that future organizations will be highly dependent on employee creativity [4]. Creativity enables organizations and their workforce to ideate, create and develop new working methods that are guided by the new digital possibilities; make use of these possibilities; and generate new working methods and practices with new, additional value propositions [7]. In this context, it is not surprising that both practitioners and scholars are engaged in searching for predictors of employee creativity [8].

Scholars from a broader range of research fields show that integrating technical and digital tools can effectively stimulate human creativity through allowing employees to use information and communication technologies (ICT) in their occupations [9,10]. For example, Zachos and coauthors [11] shows how digital support and human creativity can be used for the benefit of health and safety management. In turn, Spadotto et al. [10] show that new ICT-applications can and should be creatively used in healthcare to enable a further development of the sector. Indeed, a growing body of the new research related to this sector shows that ICT-supported creativity can enable healthcare practices improvement, innovation and operation [12,13,14].

Despite the research evidence above, there is a shared agreement among scholars that it remains unclear how different factors interact in predicting employee creativity in the workplace [8,15]. One of the main discoveries to date in this body of creativity literature is that the effects of employees’ personal characteristics on creativity are influenced by contextual factors [15,16]. For example, Joo and coauthors [17] have examined effects of such contextual factor as ‘good relations with supervisor’ and found that proactive employees can exhibit a higher level of creativity when their relationship with their supervisor is strong. In turn, Cai and coauthors [18] have focused on the interaction between the two contextual factors and found that employees with relatively high positive capital (PsyCap)—i.e., individuals with high efficacy, hope, optimism and resilience—exhibit the highest levels of creativity when the contextual factors ‘supervisor support for creativity’ and ‘job characteristics’ are both high. This research signals that both singular and multiple contextual factors can exert a significant stimulating influence on the relationship between the personal factors of employees and their creativity.

Given the diversity of possible contextual influences on employee creativity, little clarity exists regarding how different factors interact in predicting employee creativity [15,19]. One reason for this is an absence of a conceptual framework in creativity literature that would integrate and organize effects of various factors. Yet, such organization is of a need if we are to advance the field of creativity research. Such organization is also needed if we are to advance organizational practices aimed at optimizing employee creativity. In this paper we aim to contribute to filling this gap. We systematically review empirical studies on predictors of employee creativity published in the past 30 years and organize findings following an established human resources management framework: Ability–Motivation–Opportunity (AMO) [20]. We use AMO theory to categorize the predictors of employee creativity and to develop a structured scheme to explain why singular and multiple contextual factors have a distinctive influence on the relationship between personal factors and employee creativity.

AMO theory proposes that employees’ performance in organizations is guided by the presence of three work systems: *ability-enhancing practices* (e.g., using appropriate selection, hiring, and training instruments to improve employees’ abilities); *motivation-enhancing practices* (e.g., designing practices to stimulate employees’ motivation); and *opportunity-enhancing practices* (e.g., providing opportunities to enable employees to perform better). In the absence of any of these work systems in organizations, employees tend to exhibit low performance. Given that creativity is an important indicator of employee performance, we propose that the consideration of all three work systems is fundamental in explaining employee creativity. Our suggestion is that AMO theory can help categorize contextual and personal variables into three dimensions (i.e., ability-, motivation-, and opportunity-enhancing practices) and develop combination and multiplicative models that explain employee creativity.

According to the AMO literature, in the multiple configurations of bundles of the three AMO dimensions, there are two main models of interactions: the combination model (i.e., motivational- or opportunity-enhancing practices that separately activate the positive influences of ability-enhancing practices on creative performance) and the multiplicative model (i.e., motivational- or opportunity-enhancing practices that jointly activate the positive influences of ability-enhancing practices on creative performance) [21]. Acknowledging ability-enhancing practices as a prerequisite for performance [22], the existing research suggests that the combination model indicates higher performance and the multiplicative model indicates the highest performance [23]. Thus, the two main models are applied in our conceptual framework to complement the interactional perspective on creativity by clarifying how personal predictors (i.e., ability-enhancing practices) leading to a higher or the highest level of employee creativity can be explained by different types of interplay among contextual predictors (i.e., motivation- and opportunity-enhancing practices).

In summary, by employing the AMO perspective, we aim to open a new avenue that theoretically advances and refines the current understanding of interactions in pursuit of employee creativity in the digital era (i.e., the effects of contextual factors on the ‘personal factors-employee creativity’ relationship). Based on developing a human resources (HR)-relevant classification of personal and contextual predictors of employee creativity, we consider proposed synergies among various integrations of these predictors and their distinctive interactional effects on employee creativity. Extending the previous reviews on classifying creativity predictors [15,19], this work systematically analyzes how the different types of interactions towards different intensity levels of creativity can serve as a guiding framework for future research in this area and provide a basis for practitioners who want to systematically stimulate their employees’ creativity in the digital era.

We begin this paper with a definition of creativity and by presenting the theoretical lens that we use to organize our creativity research. The next section provides the applied research methodology. A rigorous systematic search is conducted to bring together the extant evidence-based empirical studies that are relevant in supporting the analysis in our current research. Thereafter, based on these selected articles, we introduce a categorization of creativity antecedents into the AMO model to describe their direct effects on creativity under different functions. Next, we explain how the two alternative models (i.e., the combinative model and the multiplicative model) can help to address interactions in creativity, and we provide some reflections on employee creativity in the digital era. We end this paper by discussing theoretical implications and future research avenues before offering practical implementations and limitations.

## 2. Theoretical Lens

### 2.1. Definition of Creativity

Creativity has been a topic of interest to both scholars and practitioners for more than 35 years. Grounded in the discipline of organizational psychology, creativity is consistently defined as the employees’ production of novel and useful products in any domain [5]. This widely accepted definition has been used as foundational in many disciplines [24]. The emphasis on creativity as an outcome, instead of the mental process through which creative ideas ultimately emerge, allows creativity to be quantified with relative ease and consensus [5,25]. Some researchers have distinguished the different types of conceptualizations of creativity theoretically [26] and empirically [27]. These scholars propose that creativity is not a uniform construct across all settings; instead, several types of creativity need to be differentiated based on the context in which they were developed. For example, Burns and Stalker [28] were the first to systematically study the differences in working environments that stimulate creative behavior among employees. Through their research they differentiated mechanistic environments that trigger routine thinking and working and so-called organic organizational structures that trigger creative working habits. Next, De Bono [29,30] was among the first scholars to distinguish creative thinking from routine-based thinking contributing therefore to understanding of the human creative potential in organizations. Based on these research findings, organizational behavior (OB) scholars have been focusing on various leadership explanations triggering employee creativity in organizations [31]. We commend these efforts and argue that they need to be complemented by a more detailed understanding of how different antecedents jointly affect employee creativity. Hence, rather than providing an exhaustive review of the employee creativity literature within a broader range of research background, we aim to take stock of research conducted in line with employee creativity concept as it is used in the OB stream of research.

Creativity can be observed at the individual, team, and organizational levels [8]. Thus, to be creative, team or individual ideas should be both novel and useful and have potential value for organizational development. Creativity can therefore involve both minor incremental adaptations and radical breakthroughs. Creativity is different from innovation. Innovation refers to the development and implementation of new ideas [32]. As individual or team creativity is often a first step to innovation, both management scholars and practitioners are looking for management tools and approaches to facilitate employee creativity and thereby obtain a better means of initiating the innovation process [8].

### 2.2. Relevance of AMO Theory to Creativity Research

AMO theory is one of the most established (HR) management theories concerning the facilitation of employee performance. The theory suggests that people perform well when they have the skills, knowledge and abilities to perform (A = abilities); when they have motivation to perform (M = motivation); and when they are provided with opportunities and support from an organization to perform (O = opportunity) [20]. The equation for this theory is *p* = *f* (Ability, Motivation, Opportunity). That is, to increase the three dimensions in pursuit of high employee performance, organizations can provide various HR practices: ability practices, motivation practices, and opportunity practices. Specifically, regarding ability practices, rigorous selection and extensive training are employed to ensure that employees have the appropriate abilities to achieve performance. Regarding motivation practices, performance appraisals, incentives and rewards, promotion, and relationships with supervisors motivate employees to enhance working behaviors. Regarding opportunity practices, flexible job design, teamwork, employee participation, organizational climate and culture for support, and information sharing empower employees to perform better [33,34,35]. The relevance of AMO theory and creativity can be explained in the following two ways.

First, as suggested by the aforementioned arguments that the HR practices within AMO theory can be applied to enhance the creativity predictors, we propose that ability practices are personal creativity factors, and motivation and opportunity practices are contextual creativity factors. In particular, ability practices (e.g., training and recruitment) enable employees’ natural capacities (e.g., skills, experience, and attitudes) that are relevant for performing tasks, which emphasizes obtaining employees with given personal attributes [36]. This point runs parallel to the focus on practical implementation in the creativity research on organizational practices that select and train employees with personal characteristics that are decisive for creativity (e.g., creative self-efficacy, and creative personality). Motivation practices include organizational incentives (e.g., pay and performance appraisal) that extrinsically motivate employees, and such practices are relevant to certain contextual creativity factors (e.g., rewards, and goal setting). Opportunity practices are based on organizational support theory, job design theories and the empowerment literature, and they include favorable employee-involvement practices for employee performance (e.g., team working, job design, and organizational (climates for) support). The above-noted practices have all been examined in the creativity literature as contextual factors [8].

Second, AMO theory suggests that the interactive relationships among ability, motivation, and opportunity practices can generate a positive synergistic effect on performance [34]. That is, employees will perform creatively when they have abilities and desirable attributes, when they are motivated adequately, and when they can fully participate in work processes. Specifically, in the combination model, motivation- or opportunity-enhancing practices in combination with ability practices contribute to higher levels of creativity while in the multiplicative model, practices from the three dimensions are presented simultaneously such that motivation- and opportunity-enhancing practices jointly support and reinforce ability-enhancing practices in pursuit of the highest level of creativity. We take the personality trait of openness—a personal predictor in the creativity literature that can be ascribed to an ability-enhancing practice according to our categorizations above—as an example to illustrate the two main interactive models. Based on the positive direct effect of employees’ openness on their creativity, scholars have found that team diversity (ascribed to an opportunity-enhancing practice because it provides more opportunities for employees to access new information) activates the influence of openness to support a higher level of creativity [37]. Furthermore, the research indicates that when feedback (a motivation-enhancing practice because it stimulates employees’ motivations) and a heuristic task (an opportunity-enhancing practice because it provides opportunities for employee engagement) are both present, employees with openness will achieve the highest level of creativity [38].

The following sections mainly reflect the two points above to analyze interactions in creativity within AMO theory. We first provide the application of AMO theory by reviewing the extant empirical studies on the antecedents of creativity and organizing the findings following AMO theory. That is, based on their different functions, we classify creativity predictors into three dimensions (i.e., ability-, motivation-, and opportunity-enhancing practices). Next, we draw on the basic types of HR practice bundles to provide a two-model framework to review how they interact to support a high level of employee creativity.

## 3. Methodology

### 3.1. Literature Search

Our literature search was limited to peer-reviewed articles published in high quality journals with a substantial impact. We searched articles in the ISI Web of Knowledge’s Social Sciences Citation Index (SSCI) database following the four steps depicted in Table 1. The inclusion and exclusion criteria were jointly decided by four of the authors.

To begin our search, we sought articles with the keyword “creativity”. The initial search resulted into more than 28,000 papers. We noted that in this sample, there were less relevant articles such as those related to library studies and sports. Considering our interest in organization and management studies with a focus on employees, we included additional criteria in the selection, for example, by narrowing the search domain, time period, and search area. Finally, we also decided to focus only on studies published in the leading management and organizational behavior journals such as *Organization Science* (OS), *Journal of Organizational Behavior* (JOB), *Journal of Management* (JM), *Academy of Management Journal* (AMJ), *Journal of Applied Psychology* (JAP), *Organizational Behavior and Human Decision Processes* (OBHDP), which thus resulted in a pool of 323 articles. We selected three general management journals and three organizational psychology oriented journals that according to Anderson et al. [8] are classified as top tier journals that publish about this topic. That these journals are top tier is confirmed by their impact factors, which are: OS: 3.257; JOB: 5.0; JM: 9.056; AMJ: 7.191; JAP: 5.067; OBHDP: 2.908.

Next, we filtered the articles based on their titles and abstracts. Specifically, given our research objective and that the majority of rigorous research that has been conducted and published in the past in this area is of a quantitative nature, we focused on including only quantitative studies. We only included studies that concern individual employee creativity rather than team creativity, because the AMO theoretical framework primarily addresses the development of outcomes at the individual employee level [36]. Moreover, based on the titles and abstracts, we also removed duplicates, editorials, news reports and review papers from the sample. As we focus on business settings and for that reason exclude all other settings, such as nursing and education.

Finally, we engaged in close reading of the full articles and selected only articles that examined the relationships between personal/contextual factors and individual creativity in organizations. Articles that focused on innovation-related behavior (IRB) and other creativity-related topics were excluded.

To complete our search, we checked the reference lists of the selected articles to search for additional highly relevant papers that did not land in our sample due to the journal selection strategy. The additional articles from other journals all met the standards of the selection criteria. The final sample of articles used in this literature review consisted of 115 papers.

### 3.2. Descriptive Results

This section describes the descriptive results of the selected articles. Figure 1 presents the growing tendency of the creativity research during the last 30 years. Since the 1990s, a growing body of academic work on creativity, especially from 2000, 2009, and 2015, has witnessed the importance of creativity in the organizational research area. Creativity has been studied in a number of fields within the social sciences such as organizational management, psychology, business economics, behavioral science, art, and educational research. Given our AMO perspective, we focus on the management research field and on key journals in this field. The significant growth of academic research on creativity also indicates that most industries require employees to be creative by integrating digital and twenty-first century skills [7]. 

## 4. Application of AMO Theory

As the aforementioned arguments indicated, the personal and contextual predictors reflect practical implications for HRM practices. Therefore, to categorize predictors of employee creativity, we classified them into the three dimensions of the AMO model, which is a new approach to understanding employee creativity in depth. Specifically, the personal factors of employee creativity were categorized into ability-enhancing practices while the contextual factors were classified into two dimensions—motivation- and opportunity-enhancing practices—according to their different human resource management (HRM) functions (more theoretical explanations on the categorizations are provided below).

This classification requests further explanations. Although AMO theory suggests that employee selection and training are the two main aspects of ability-enhancing practices [20,34], few creativity studies address these two facets at the same time. Therefore, to categories the predictors of employee creativity under the ability dimension, in line with the creativity literature, we refer to personal characteristics as ability-enhancing practices. This is because HR selects individuals based on specific personal characteristics and provides specific training to encourage employees to use these personal characteristics in pursuit of their creativity. For instance, Hirst and coauthors [39] examined the positive role of learning goal orientation on creativity and suggested that selecting learning-oriented candidates is likely to boost creative performance.

However, given that the predictors of employees creativity are multidisciplinary and conceptualized differently in various research contexts [40], categorizing them using the AMO framework is not always easy. For example, although some researchers claim that job design motivates employees to engage in creativity [41], in line with AMO, we classify it as a typical opportunity-enhancement practice [20] because job design creates opportunities that either enable or constrain employees’ motivation to pursue creativity. For example, scholars have shown through designing high autonomous jobs employees feel they have more opportunities to participate [23,33] and, therefore, to engage in creative performance. In this study, we mainly focus on the theoretical definitions and functions in the AMO literature to categorize the three dimensions of the factors of creativity.

### 4.1. Predictors of Creativity: Ability-Enhancing Practices

Ability in AMO theory stresses that HRM should design selection and training procedures to acquire talent with personal characteristics that support performance [20,33]. Especially in the new digital era, employees’ abilities to deal with unexpected challenges and then produce creative solutions are highly required [4]. As mentioned above, ability-enhancing practices emphasize the idea that employees with certain personal attributes such as a creative personality, self-efficacy and intrinsic motivation may facilitate creative outcomes. We follow the existing reviews [8,15] to divide ability-oriented personal factors into four aspects: personality/cognition, personal motivations (different from the motivation-enhancing practices that suggest contextual factors motivate employees, the personal motivation in ability-enhancing practices is about internalized personal characteristics), and affect/mood. The following review of different types of creativity also refers to these four aspects.

*Personality/cognition*. Most of the research shows that employees with certain dispositional differences such as a creative personality, a big-five personality, and divergent cognitions may produce more creativity. A theoretical explanation is that these personalities encompass the general tendency to seek challenges and produce creative ideas. For example, Baer [37] found a positive relationship between openness to experience and creativity. Moreover, an emerging stream of research has recently captured the benefits of proactive personalities and behaviors for creativity because proactivity initiates changes to stimulate individuals’ gaining of knowledge and achieving creative goals [42]. Regarding the benefits of cognitions, Tierney and coauthors [43] highlighted the importance of cognitive style by illustrating the positive influences of innovative cognitions on creativity.

*Personal motivation*. A few studies based on Amabile’s [5] componential theory examine the beneficial role of *intrinsic motivation* (defined as individual engagement for interests and internal satisfaction) in predicting creativity [44]. Similarly, several related constructs such as intrinsic interest and prosocial motivation have a positive association with creativity [45]. When employees are excited and fully engaged in tasks, they tend to explore creative ideas and methods. In addition, in *goal orientation* studies, Gong and coauthors [46] found that learning orientation is associated with creativity because it supports individual development and competence, leading to creative achievements. In addition, most of the research is now examining creative self-efficacy as a significant predictor of creativity [47]. In addition, some self-centered predictors can also act as a motivational driver for creativity, which emphasizes the personal view of the self. For example, Jaussi and coauthors [48] suggest that creative personal identity predicts creativity because this self-definition helps individuals realize and utilize their unique internal properties (e.g., intelligence) to engage more in creativity.

*Affect/mood*. Some research investigates the link between emotional phenomena such as *affects*, *mood*, *and emotion*, and employee creativity because these attributes are directly related to people’s actions on the job. Specifically, emotions exert significant effects on the processing of information, which builds and broadens individuals’ cognitions and resources. For example, Bledow and coauthor [49] found that a shift in the experiencing of affects (the initial experience of both positive and negative affect followed by decreasing negative affect and increasing positive affect throughout a short time frame) is associated with individual creativity.

*Others*. A few interesting studies consider the influences of other personal-centered determinants such as perspective taking, autonomy orientation, focus of attention, creative ability, and beliefs. For example, Yu and Frenkel [50] found that employees’ feeling of obligation to act appropriately is a superior predictor of creativity.

### 4.2. Predictors of Creativity: Motivation-Enhancing Practices

Motivation in AMO theory focuses on using organizational compensation and incentives to extrinsically motivate employees to use their skills, knowledge and enthusiasm to achieve higher job performance [20,51]. Typical practices include developmental performance management, incentives and rewards, promotion and career development, and job security. These practices express organizational respect to workers and motivate them to achieve desired job performance. Appelbaum et al. [20] stressed the importance of developing a mutual trust relationship and supervisory behaviors among motivation-enhancing practices because these issues function to encourage individual engagement. In our study, we suggest that goals and some leadership issues are also motivation-oriented practices. Specifically, goal setting, as an effective motivational technique, provides clear targets and directs individual attention towards task achievement [52]. Leadership styles and behaviors, such as the motivation-promotion supervisory style, directly guide employees towards creative outcomes through encouragement and modeling [53]. It should be noted that, given the different functions of leadership styles, we suggest that empowering leadership is an opportunity-enhancing practice because it focuses on providing employees opportunities for authorization to achieve effective outcomes [54].

*Rewards*. In the HR literature, rewards are identified as a positive and typical motivation-enhancing practice [20,34]. The research on the influences of extrinsic rewards on creativity maintain two opposing ideas: some researchers have found that rewards contribute to creativity [55] while others have found a negative relationship [5]. From a cognitive perspective, the argument for the negative effects of rewards on creativity originates from the assumption that rewards—as an extrinsic motivation—constrain individual cognitions and reduce self-interest to undermine creativity. From a behavioral perspective, researchers claim that rewards fulfill employees’ need for competence, which supports their creative efforts [44,56]. In an HRM context, rewards are an important aspect of payment systems and signal the behaviors and outcomes that are expected in the organization, thereby incentivizing employees toward good performance (e.g., creativity).

*Leadership and supervisory behaviors*. Most of the research has examined the role of various *leadership styles and supervisory behaviors* [8] because they send informational signals that serve as motivational stimulation [5] such as *transformational leadership*, *aversive leadership,* and *benevolent leadership* [43,57]. With regard to leadership, leader behaviors such as supervisor support (for creativity) and unconventional behaviors have also been found to affect followers’ creativity [18,42]. Furthermore, leader-member relationships, especially *leader-member exchange (LMX)*, *trust,* and *justice*, relate to creative performance [58,59].

*Goals and expectations*. One of the most salient factors for creativity is goal setting, which motivates employees by affecting self-regulatory mechanisms [60]: goals clarify targets and requirements, leading individuals to judge their behaviors and then direct their attention to facilitate creativity. Shalley’s [60] study underscores that original and productive goals facilitate creativity. The research has also shown that work expectations lead to creativity by shaping employee responses to the realization of various potential outcomes.

### 4.3. Predictors of Creativity: Opportunity-Enhancing Practices

Organizing work processes that ensure that employees have opportunities to display their skills and motivations is the main task of opportunity-enhancing practices [20]. The basic assumption involves a positive link between opportunity-enhancing predictors and creativity such that when an organizational structure provides wider participation for employees (e.g., horizontal organization), they will regulate their behaviors on creative tasks [51]. First, well-designed tasks (e.g., job autonomy) nurture employees’ problem-solving responsibilities to inspire creativity [61]. Second, workers require more support from organizations [50,61], as reflected by the organizational climate and culture (e.g., resources, support for creativity) [62]. In the HR literature, this aspect refers to individuals’ recognition of an organization’s attributes (e.g., practices, policies and procedures). This line of research indicates the importance of building a work environment where employees have the opportunity to interact, which is categorized as an opportunity-enhancing practice. Furthermore, relevant teamwork attributes offer a desirable platform for employees to access new ideas and perspectives on creativity [39]. Finally, the research drawing on social network theories has addressed the influences of structural properties that reflect the social relationships between individuals and organizations or environments [63].

*Job/task design*. The importance of job design lies in allowing changes for individuals to use their personal abilities and knowledge to engage in creative activities. Regarding job autonomy, Liu et al. [64] examined the role of autonomy such that autonomy directly promotes creativity. Considering the benefits of feedback on creativity, Zhou [65] found that employee creativity is generated by the receipt of positive feedback delivered in an informational style. Moreover, other aspects of work design, such as time/performance pressure, task structure and routinization, have been found to influence creativity [41,61,66].

*Organizational climate and culture*. A number of studies have made substantial progress in facilitating our understanding of how organizational climate and culture exert influence on employee creativity, especially the type of organizational support that guarantees employees’ perceptions of organizational assistance and encouragement [44]. For example, De Stobbeleir [67] tested whether perceived organizational support for creativity had a positive effect on creativity.

*Teamwork*. Since creativity requires a platform to exchange and obtain information [60], teamwork, which is identified as the *presence of coworker (behaviors)*, *information sharing/exchange*, *team/organizational diversity*, and *team learning behaviors* [54,60,66], also benefits individual creativity. For example, Khazanchi and Masterson [59] found a positive association between information sharing and creativity. In addition, some research has provided evidence for an understanding of the role of coworkers, who are found to act as role models and provide new sources of information to support creativity. An example is a study by Zhou [68] that showed the positive relationship between coworkers’ external networks and employee creativity.

*Social network*. Social networks indicate an individual’s access and control within and outside the organization. The context of social relationships explains the influence of relationships between employees and their surroundings on creative outcomes [63]. This attribute reflects organizational opportunities for employees’ engagement in tasks [20]. Specifically, in the field of creativity, network ties and network structure and position are the main dimensions [63]. Sosa [69] examined network interactions and structures to show that the strength of a dyadic relationship (to the extent that it captures the work-related closeness of the interacting actors) positively stimulated creativity.

The above re-categorized factors of creativity from the existing empirical studies substantially link AMO theory with the creativity literature. Based on this finding, in the following section, we present further theoretical explanations that support the framing of the complex interactions in the creativity literature based on bundled practices within the AMO framework.

## 5. New Typology of Interactions

As noted above, our understanding is limited because of the few mixed findings in the creativity research (i.e., how the different types of interactions may predict different intensity levels of creative performance) [15], and practitioners are thus ineffectively guided in managing employee creativity towards a higher or even the highest level of creativity. Accordingly, to clarify these essential problems, in this section, our analysis is based on the above categorizations of creativity predictors to develop a new framework that displays the variety of interactions in the creativity literature. Scholars in the AMO theory research field have suggested two main configurations for bundles of practices towards (creative) performance—the combination model and the multiplicative model [21]. That is, in the two main models, the practices are interdependently aligned to generate performance. Moreover, as new-era workplace characteristics emphasize that organizations should continually compete for the best talent, the occurrence of various bundles of HRM can be implemented towards new forms of work in the digital era [70]. Thus, our proposed new framework includes the two main models to explain how ability, motivation, and opportunity predictors interact in different ways to predict employee creativity. Notably, given our basic assumption that contextual factors moderate the relationship between personal factors and creativity, the influence of ability predictors on creativity can be accentuated by motivational and opportunity practices both separately and jointly. Figure 2 represents the framework including the two main interactive models.

### 5.1. The Combination Model

The combination model, referring to *p* = *f* [A(1 + M + O)], suggests that ability is a prerequisite for performance and that motivation and opportunity can separately help in the presence of sufficient ability [21]. That is, the motivation or opportunity-oriented creativity predictors accentuate the effects of the ability-oriented predictors on creative performance. Specifically, acknowledging that employees with abilities (e.g., creative personality and positive mood) can effectively contribute to creative performance, organizations should make good use of employees’ personal benefits by displaying two dimensions of practice. The first dimension includes motivation-enhancing predictors that motivate employees, which may generate additive effects to employee creativity while the second includes opportunity practices that provide opportunities to engage in creative endeavors, which may facilitate the incorporation of employees’ skills and abilities. Thus, in this section, we organize our creativity research to propose two types of combination models to reflect the interactive influences in the creativity literature—the interactive effects of ability and motivation (A × M) and of ability and opportunity (A × O) (depicted as Combination Models (a) and (b) in Figure 2).

*P = f (A × M)*. Tierney et al. [43] studied the relationship between cognitive style and employee creativity by considering the moderating role of LMX. Cognitive adaptors perform more creatively when they work within a high-quality LMX dyad rather than within a low-quality dyad. Moreover, Qu et al. [71] found an interactive relationship between relational identity and leader creativity expectations, showing that employee identification yields a significant increase in creativity when leaders provided expectations of creativity. Considering the effects of mood, George and Zhou [58] found that a negative mood was significantly and positively associated with creativity when perceived recognition and rewards for creative performance and clarity of feelings were high [72] or when interactional justice or trust relationships were high [58]. Some research has explored the associations between other ability-enhancing predictors and creativity. Chen et al. [73] stated that the positive influence of individual initiative and skill variety on creativity were stronger when employees were provided with adequate creative resources. In three studies, Aleksic and coauthors [74] found that clear goals positively moderated the relationship between personal preference for creativity and creativity.

*P = f (A × O)*. There is considerable research examining the effect of work characteristics (i.e., opportunity-enhancing predictors) on the relationships among personality, cognitions and creativity [75]. For example, Zhou and Oldham [38] focused on the creative personality interacting with expected developmental assessment strategies and found that individuals with creative personalities exhibited creativity when they expected to be self-administered. To establish the underlying goal orientations-creativity relationship, Hirst and colleagues conducted two studies to display the moderation of different opportunity-enhancing predictors. They found that employees’ learning orientation contributed to a higher level of creativity when team learning behavior was high [39] and when centralization was low [51]. Regarding identity, Yoshida et al. [76] found positive moderation effects of organizational innovative climate on employees’ identification and their creativity. Researchers have provided evidence on how task attributes activate the linkage between mood and creativity. For example, Binnewies and Wörnlein [77] examined the effect of job control, showing that a high level of job control amplified the desirable influence of positive effects on creativity. A few studies have considered the effects of voice behaviors on creativity and found that the positive effect of employees’ voice behavior on creativity was stronger when they worked in a more innovative climate [78].

This set of studies reflects the different functions of motivation- and opportunity-oriented factors in activating the relationships between ability-oriented predictors and employee creativity. Generally, the results of creativity seem to be higher when motivation- or opportunity-enhancing practices are involved in the main influences of ability-enhancing practices on creativity. Take identification as an example. Two studies from Qu et al. [71] and Yoshida et al. [76] examined the moderating effects of leader creativity expectations and support for innovation, respectively. Their findings showed a significant enhancement of motivation- and opportunity-oriented variables, with more variance in creativity explained by the variables of leader creativity expectations (R^2^ = 0.15) and support for innovation (Pseudo R^2^ = 0.05). Moreover, among ability-oriented factors, unlike stable personality attributes, some psychological characteristics (e.g., cognitive styles and affects) are more subject to variation. Thus, the benefits of their potential positive influence on creativity should be contingent on the augmenting or substitution effects of motivation- or opportunity-oriented practices. This idea aligns with the contextual perspective of mood–creativity relationships, which is that the creativity advantages of positive or negative moods should be considered in a task environment where designing tasks varies in terms of time plans and information orientations. More notably, the combination effects on employee creativity can be explained from the perspective of digitization. Specifically, the digital age provides new enabling factors for generating and sharing information or knowledge that could dramatically impact employee outcomes [4]; therefore, when employees with knowledge of utilizing technologies are highly motivated or provided opportunities to utilize such tools and technologies in the workplace, they become more digitally engaged in solving problems in creative ways.

### 5.2. The Multiplicative Model

Since ability-, motivation-, and opportunity predictors are likely to codetermine (creative) performance, scholars present a multiplicative model to analyze how the three dimensions of predictors jointly contribute to the prediction of the highest level of creativity. Specifically, the multiplicative model illustrates the traditional argument in AMO theory, which is that ability, motivation and opportunity together operate in a complementary or interactive manner: *p* = *f* (A × M × O) [40] such as by exerting a larger effect and compensating for the disadvantageous influences of other factors. Considering the enhancements of several motivation and opportunity predictors separately in terms of the above-noted ability-creativity relationships (i.e., combination model), the multiplicative model emphasizes the joint effect of predictors from the ability, motivation and opportunity dimensions simultaneously. The multiplicative model provides a more comprehensive picture of interplay in creativity. Scholars have found that performance is highest when organizations simultaneously invest in enhancing employees’ ability, motivation, and opportunity. Considering the significance of boosting employee creativity to the highest level [8], the multiplicative model provides a useful and practical perspective for creativity scholars to consider the three factors simultaneously—with each factor supporting the other two—operating as synergistic bundles to achieve the highest level of creativity [36]. Among the selected articles, moderation generally accounts for the three-way interactions. This model is depicted as the Multiplicative Model in Figure 2.

Baer et al. [79] found that extrinsic rewards, together with working on complex jobs, exerted significant and positive moderating effects to stimulate employees with an adaptive cognitive style to produce creativity. Kim et al. [42] studied the potential positive role of a proactive personality and showed that a significantly high level of creativity was achieved when proactive employees received supervisor support for creativity and a requirement for high job creativity.

Considering the advantages of personal motivation in predicting creative performance, research has examined the conditions under which these motivations could be most explanatory for employee creativity [61]. For example, Zhou [68] examined the joint condition of supervisor developmental feedback and the presence of creative coworkers on the creative personality-creativity association. Two studies confirmed that employees who have less creative personalities may exhibit creativity when creative coworkers are present and when supervisors provide developmental feedback. Moreover, De Clercq et al. [80] found that the joint effects of learning orientation, high goal congruence, and low task conflict contribute to the highest level of creativity.

With regard to multiplicative effects, including other aspects of ability-enhancing predictors, Zhang and Zhou [67] conducted two studies on the interaction among uncertainty avoidance, trust, and empowering leadership that affects creativity such that the highest creativity could be achieved when employees had high levels of uncertainty avoidance and trust in their supervisors and when leaders displayed empowering leadership. In addition, Zhou et al. [68] proposed a three-way interaction among employees’ perception, supervisor support for creativity, and access to resources, showing that opportunity perception had the strongest positive relationship with employee self-perceived creativity when supervisor support for creativity and access to resources were both high.

The empirical evidence seems to suggest that the multiplicative model of interactions predicts the highest level of employee creativity. For example, as researchers have found that openness to experience is important for creativity, studies provide greater evidence that openness can be activated by motivation- and opportunity-enhancing practices simultaneously. George and Zhou [69] showed that openness to experience would result in high levels of creative behavior if feedback valence was positive and job holders were presented with a heuristic task that allowed them to be creative for jobs with unclear ends (changed R^2^ = 0.04) and with unclear means (changed R^2^ = 0.04). Moreover, the findings for some potential negative ability-oriented variables may even predict creativity when suitable motivation and opportunity practices are properly combined. Baer et al. [79] explored the interaction between personal cognitive styles, rewards, and job complexity as it relates to differences in creativity: employees with an adaptive cognition exhibited higher creativity when they worked on simple jobs and received increased rewards. The results highlight that creative potential is not uniform and can be shifted by contextual motivations and opportunity benefits. At the same time, together with the findings in the combination model [58], some evidence from potential negative predictors, such as rewards, may also act as an enhancement to generate creative outcomes in specific circumstances. A prominent precondition is the task characteristics that provide employee benefits in the context of a good “person-job match”.

Both these arguments on the two main interactions set forth above are highly consistent with the basic AMO assumption that ability, motivation and opportunity practices are synergistically bundled towards performance [20,21]. That is, operating as different functions, the three dimensions of creativity predictors (i.e., ability-, motivation-, and opportunity-enhancing creativity factors) interact with one another, with one factor supporting the other two. To illustrate, the combination model generates a higher level of creativity than the main effects of ability predictors on creativity, and the multiplicative model contributes to the highest creativity. The extent to which ability-oriented predictors facilitate creative performance depends on the bundles of motivation- and opportunity-oriented presiders as weakness in any of the three may reduce creativity. A specific configuration (i.e., motivation- and opportunity-enhancing variables together bringing out the creative potential of the ability-enhancing variables to achieve the highest level of creativity) indicates that “changing one leg without careful consideration of the other two is typically a mistake” ([81], p. 325). Furthermore, digitization gives organizations a unique opportunity to bring their HR function to a new level, and multiplicative bundled HR practices can enable employees to be most effective towards creative achievements. Specifically, when employees are taking training courses to adjust to digitized working tasks, they are not only eager to be motivated (e.g., leaders encourage their followers’ digital initiatives), but they also seek out opportunities (e.g., appointing a technological task fit their digital excellence) to further boost their creative endeavors.

## 6. Discussion

The rapid digital transformation of the past years has increased our awareness of the importance of creativity in modern organizations. Therefore, it is urgent to have an increasingly clear understanding of how digital technologies can act as a promotor of employee creativity in various industries. Specifically, prior studies have demonstrated that ICT can support and trigger employee creativity in a number of industries [10], and specifically in the healthcare sector [13,14]. Beyond ICT-oriented industries, less ICT-oriented organizations and business settings also require employees to perform creatively and can use ICT to trigger this. The predictors of creativity have been investigated since 1990s where digitalization began to take shape all over the world. Yet, only recently scholars have become aware that these predictors should be re-identified in accordance with the HR practices. This means creative predictors should be distinguished in line with an HRM theoretical framework. This approach will enable researchers and practitioners to develop a better understanding on how different predictors can jointly facilitate employee creativity to the highest extent.

Given the importance of the under-developed interactional perspective on creativity in the context of the digital era, we believe it is time to draw a comprehensive picture. Drawing on the AMO framework, we provide a new approach to the investigation of employee creativity from the HRM perspective. Specifically, we refine the categorization of creativity predictors into three dimensions (i.e., ability-, motivation- and opportunity-enhancing practices). In doing so, we extend the current understanding of the HR-oriented characteristics of creativity predictors. Based on the basic assumption that person-creativity relationships can be augmented by contexts with different functions, in this paper we presented two main models of interactions (i.e., the combination model and the multiplicative model). Our conceptualization helps to enrich the limited interactional research by shedding light on the various functions of creativity predictors and extending scholarly understanding of the different types of interactions in the creativity literature. As such, we address this research gap and answer scholars’ call for systematically exploring the extent to which the various interactions among predictors may maximize employee creativity [15]. Our hope is that our efforts will help move the research on this complex interactional perspective forward in new and exciting directions. For example, future research would benefit from our theoretical point to propose and expect why and how predictors and/or AMO practices may stimulate employee creativity to a different extent. We also hope that our findings can be practically implemented in organizations and workplaces, which are subject to ongoing and far reaching changes due to continuous digitization and are thus among the most significant and important practical settings.

Specifically, in this paper we show that motivation-oriented factors act as motivational stimulation to externally boost employee utilization of their abilities, skills and personal attributes to engage in creative endeavors. This idea complements the basic motivational mechanisms through which contextual factors stimulate employees’ internal motivations to exhibit creativity [47] by showing that the motivational factors in organizations enable the expression of ability-oriented facilitators. In the same vein, opportunity-oriented factors emphasizing the provision of avenues for employees to express creativity can highlight opportunities that can be performed creatively. Theoretically, when organizational inputs (e.g., providing opportunities) target employees’ personal development, employees are likely to reciprocate in terms of desirable outcomes such as engaging in creative endeavors by utilizing personal advantages [71]. In turn, if we are to reflect on the specific HRM practices in the digital era, it is clear that ability-enhancing practices emphasize employees’ acquisition of digital skills and abilities. For organizations, this often means taking away restrictions and separations among departments and creating open culture in which employee can be creative in doing their digital and Internet-based work.

By framing two main models of interactions in the creativity literature, we structure the potential bundles of ability-, motivation-, and opportunity-enhancing predictors. This new perspective is consistent with the extensive literature on person-environment fit theory, which argues that individual performance and behaviors result from the adjustment between employees and their work environments. Specifically, the compatibility between personal and environmental attributes suggests that contextual components of creativity with different effects (i.e., motivational or opportunity providing) may activate employees to adapt to creative achievements. Meanwhile, the results of our study align with the bundle perspective in AMO theory, which is that the optimal choice of HRM practices produces various results. As ability practices are considered to exert the main positive effect on creativity, motivation and opportunity practices separately accentuate creativity to a higher level and jointly to the highest level. Clearly, this bundling or synergy effect of interplay provides explanations for scholars’ arguments that the interactional influences of personal-contextual characteristics on creativity are contingent on other contextual facilitators [82].

### 6.1. Practical Implications 

As we mention above, we take an explicitly managerial perspective to assume that HRM practices are highly correlated with creativity predictors. Thus, our results also provide some practical contributions to digitizing organizations that are in the process of managing employee creativity. First, the three HRM dimensions (i.e., ability-, motivation, and opportunity-oriented practices) are definitely related to employees’ creative performance. This finding suggests that digitizing organizations should provide creativity-facilitating practices along the three dimensions. For example, such organizations must focus on selecting candidates with proper personal characteristics (e.g., creative personalities and positive psychological attributes) and designing innovative tasks. In addition, the interactional approach in favor of joint practices for HRM departments emphasizes that bundles of practices should include various functions and should be based on the principle of matching. For example, a specific effective HRM practice for employees with innovative cognitive styles is that organizations should provide non-rewarded and innovative digitized tasks instead of extrinsic rewards and simple digital jobs. Third, given that some motivation- and opportunity-enhancing practices are displayed by leaders and teams, organizations should provide training for leaders and employees. Training for managers should include setting goals, building trust and justice, and supporting employee creativity, all of which fits with a continuously developing digitized work environment whereas, for employees, training should include sharing knowledge and information and developing learning behaviors concerning changing roles and tasks due to ongoing digitization. The final implication concerns synthetically combining and enacting ability-, motivation-, and opportunity-enhancing practices to enable employee to perform creatively. For example, HR should select candidates with digital orientations, encourage them to utilize technological information resources, and build a supportive work environment to facilitate interactions and communications via digital method among employees.

### 6.2. Limitations and Future Research

This paper is not without limitations. First, some studies have investigated the interactions that primarily focus on two motivation- or opportunity-enhancing predictors moderating the ability-creativity link [39], which generate a higher level of creativity. These examinations are not contrary to our conceptual framework as they also consider the different roles and functions of motivation or opportunity factors. Future work will need to follow our conceptualizations to compare these studies and deepen our knowledge of the complex and various interactions. Second, our work includes fewer interactions involving negative creative results. For example, in George and Zhou’s study [72], positive mood was found to be negatively related to creative performance when perceived recognition and rewards for creative performance and clarity of feelings were both high. Despite our basic argument that favors the profitable interactions of predictors fostering creative results, it would be valuable for future research to consider the dark side of predictors, especially the dark side of organizational practices. For example, positive potential predictors may jointly result in a too-much-of-a-good-thing effect on creativity. Third, some of the articles referenced above are organized across different levels of analysis, yet we pointed less to the recent growing research focused on the multilevel approach. Although our framework applies to multilevel or cross-level studies, it would be beneficial for future research to determine how the various influences of motivation- and opportunity-oriented variables from different levels interact with individual-level ability-oriented variables to cause employees to perform more creatively. Moreover, although it is acknowledged that digital possibilities have been challenging traditional ways of providing HRM services within organizations, the organizational behavioral research in this specific area has received relatively less attention. Specifically, in our sample of reviewed papers, we found no example of specific digital-oriented predictors towards employee creativity. Given that creativity is situated as an economically and commercially valuable set of personal dispositions and skills in the digital age and more lately in an explosion of creativity situated as an economically and commercially valuable set of personal dispositions and skills in the digital age, it is thus necessary for scholars in the future to shift attention to the impact of digitized HRM activities on employee creative performance in a digitizing workplace. With a specific focus on the organizational behavior and HRM research areas, future research is encouraged to address the topics of digitization and its impact on workplace creativity (e.g., how digital tools influence human creativity) and creativity in other settings (e.g., creativity in healthcare). The next limitation is about the definition of creativity, as we focused only on the OB research domain and used the definition of Amabile. Although it was important for this paper to address creativity as an outcome variable, this approach limits understanding of creativity as an employee process. Given that the focus of this paper was to explore predictors of employee creativity, the chosen definition is appropriate. Yet, we suggest that future research should consider adopting broader definitions of creativity. Furthermore, our focus on understanding the interactional effects of predictors on employee creativity could have been potentially restrictive toward understanding a full spectrum of interactions in predicting employee creativity. We would like to encourage research to use other theoretical frameworks to explore and explain interactional effects of various factors on employee creativity, thus allowing for further discoveries in this research area.

## Figures and Tables

**Figure 1 ijerph-17-01038-f001:**
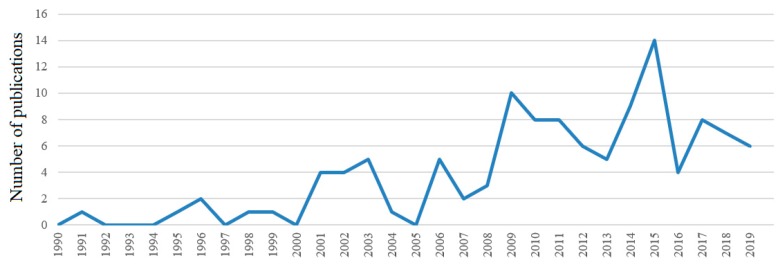
Growth in published papers on employee creativity.

**Figure 2 ijerph-17-01038-f002:**
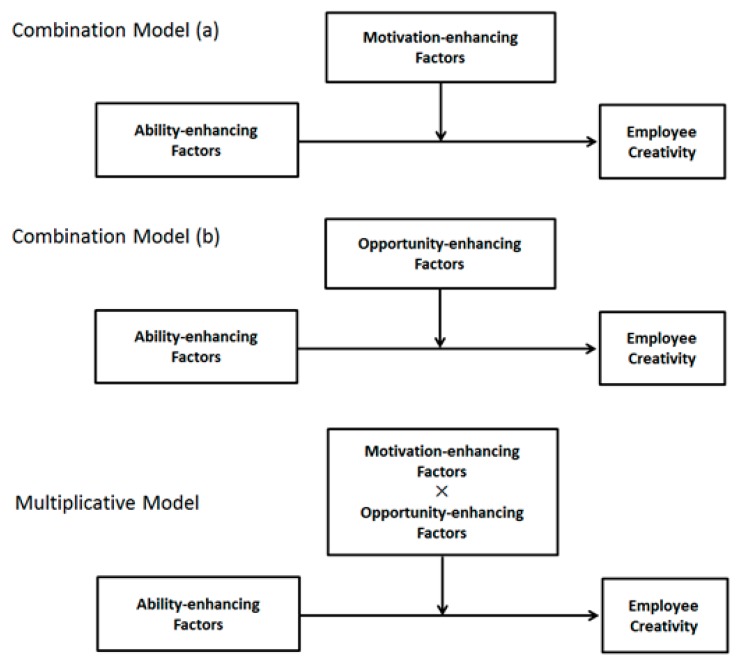
Combination and multiplicative models.

**Table 1 ijerph-17-01038-t001:** Literature Collection Stages.

Stage	Details	Number of Records
Stage 1: Keyword search	Search topic: key word “creativity”	28,091
	Time filter: Published in the years 1990–2019 (July)	26,129
	Search domain: social science	19,631
	Search area: psychology, business economics, behavioral sciences	13,173
	Duplicate records, Editorials, News reports and Review papers	11,018
	*Source title: Organization Science* (OS), *Journal of Organizational Behavior* (JOB), *Journal of Management* (JM), *Academy of Management Journal* (AMJ), *Journal of Applied Psychology* (JAP), *Organizational Behavior and Human Decision Processes* (OBHDP)	323
Stage 2: Select and Sort (based on analysis of title/ abstract)	Exclusion criteria: - Qualitative research - Studies on team or organizational creativity and innovation - Studies on the educational research and other research field (e.g., nursing)	115
Stage 3: Refined select and sort (based on analysis of article content and abstract)	Exclusion criteria: - Studies on team or organizational creativity and innovation - Studies on other research issues (e.g., IRB) - Papers not focus on influential factors of individual creativity	103
Stage 4: final selection	Additional articles: Articles by reference check should meet standards of perceived quality of rigor, relevance and readability with high citation from other journals (*n*=15)	118

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
