# Peer review of "Optimizing Employee Creativity in the Digital Era: Uncovering the Interactional Effects of Abilities, Motivations, and Opportunities"

_ijerph, 2020, doi:10.3390/ijerph17031038_

Round 1
Reviewer 1 Report
Dear Authors,
this is a very interesting and well written article which definitely contributes to the discourse on creativity! Well organizaed and structured it provides valuable understanding of this important and timely issue.
I have only two minor concerns both related to the methodology:
Please explain a little more detailed the reason for your exclusion criteria between stage 1 and stage 2 Please explain more in detail and also providing some data how exactly you decided which are the "leading management and organizational behavior journals" (page 5 line 189...).
I recommend publication after you dealt with these two minor issues.
Author Response
Reviewer 1:
Dear Authors,
this is a very interesting and well written article which definitely contributes to the discourse on creativity! Well organizaed and structured it provides valuable understanding of this important and timely issue.
Response: Thank you very much for your positive and encouraging comments on our manuscript. Your insightful guidance has helped us during our revision. We have revised our paper in accordance with your comments. We hope that you will find our revision addressing your concerns. Below are our point-by-point responses to your concerns
I have only two minor concerns both related to the methodology:
Please explain a little more detailed the reason for your exclusion criteria between stage 1 and stage 2
Response: Thank you for this advice. We added more detailed information about the exclusion criteria between stage 1 and stage 2 in the text (p. 6, line 225-237). Specifically, we aim to analyze the changes of employee creativity through providing the detailed date (e.g., coefficient, and R2) after reviewing exiting papers (p. 12, line 487-488), which provides accumulating evidence; thus, selecting quantitative papers fits our pre-specified eligibility criteria. That is, we first explained that “since the majority of rigorous research that has been conducted and published in the past in this area is of a quantitative nature” (p. 6, line 231-232), we only include quantitative studies in our review paper.
Second, we inserted that we exclude team and organizational creativity “because the AMO theoretical framework primarily addresses the development of employee outcomes (Kim, Pathak, & Werner, 2015)” (p. 6, line 234-235).
Third, we specifically highlighted that “We focus on business settings and for that reason exclude all other settings, such as nursing and education” (p. 6, line 236-237) to further help readers understand our research scope.
Please explain more in detail and also providing some data how exactly you decided which are the "leading management and organizational behavior journals" (page 5 line 189...).
Response: Thank you for this comment which motivated us to provide a stronger justification of the selected papers from such leading journals (p. 6, line 225-229). Specifically, “We selected three general management journals and three organizational psychology oriented journals that according to Anderson et al. (2014) are classified as top tier journals that publish about this topic. That these journals are top tier is confirmed by their impact factors, which are: OS: 3.257; JOB: 5.0; JM: 9.056; AMJ: 7.191; JAP: 5.067; OBHDP: 2.908.”
I recommend publication after you dealt with these two minor issues.
Response: We would like to thank you once again for your insights. They definitely helped us with our revision. We are glad that you like and provide strong support for our paper. We hope you find our current version much improved.
Reviewer 2 Report
The submission is ambitious, seeking to understand and synthesise what is known about the interactional effects of contextual and personal factors in employee creativity, and based on the understanding, report new areas for research. The research literature in different disciplines on factors that impact on human creativity including creativity in work-related activities is very large, but a systematic review and analysis can be of interest.
One strength of the submission is its systematic approach. It reports a systematic literature review, then uses the established and cited AMO theory to categorise the relevant literatures according to the different factors that can separately and jointly influence human creativity.
However, the submission also has a number of weaknesses that undermine the case for publication in the journal. One is the fit with the journal title – Environmental Research and Public Health – and the theme of Digital Health. Nowhere in the submission is there a clear link to creativity in healthcare, how digital technologies evolve work so that it might be more creative, or creative in different ways, or how digital tools can be used to enhance human creativity. Relevant work on creativity in healthcare is not cited, and research exploring how digital tools influence human creativity in conferences such as ACM Creativity & Cognition is not cited. Instead, the submission appeared to be written for more general HRM and business studies title. Instead, a more explicit fit of the research to the journal needs to be made – digitization and its impact on workplace creativity is great, and something worth researching, but it needs to recognise and reference relevant previous research.
Another weakness is the definition of creativity, which fails to build on established refinements of novel and valuable. For example, does the research focus on what Kaufman and Beghetto call Pro-C creativity, and/or does it refer to little-c creativity, or other types? More effective scoping and definition of the central phenomenon under investigation was needed. And given the focus on employee creativity, there was no attempt to differentiate between different types of employee and work role. I might expect to see some differences between roles such as production line assemblage and digital designers or architects. I accept that a detailed analysis of studies by specific roles was not feasible, but a simple separation between roles specific to the creative industries or not, or the level of education needed, might have led to more inciteful findings.
The submission also treated creativity as unitary concept, which it is not. Research in fields not cited in this paper demonstrates that different basic stages of creative processes (e.g. divergence versus convergence in idea generation) impacted by different individual and contextual factors. The unnecessarily simple definition of creativity adopted for the review limits the rigour of the analysis because it is too coarse-grain. Furthermore, it was not clear to me why team creativity was outside of the scope of the research, given that this how much creativity in the workplace happens.
In light of these criticisms it was possible to argue that literature review was too narrow and restricted at only 115 from specific journals in the business (but not health or digital disciplines). This amounts to a relatively small number of publications each year since 1990. Of course, a systematic analysis of 115 journal papers is still a substantial effort, and is applauded. However, the number and focus of the papers was only a small fragment of the research reported about human creativity and factors that influence it. Not only are there other sources of research that investigate factors that influence employee creativity, but also there are a larger set of studies that provide evidence towards theories and models of personal and contextual factors that influence human creativity. For example, the 2008 meta-analysis of contextual factors on human creativity by Matthijs Baas and colleagues drew on over 160 papers and revealed patterns about the nature of the space and environment in which people are creating. This clearly has some import to the research? And again, the literature review does not reference digital health or health explicitly.
The resulting models from the analysis of the literature are relatively simple, and most of the reported analysis returned to instances of cited literature categorised according to the AMO theory. Given my concerns about the source literature used, I struggled to see how the work reported in the paper could provide meaningful guidance to organisations in employee hiring and work redesign.
Author Response
Reviewer 2:
The submission is ambitious, seeking to understand and synthesise what is known about the interactional effects of contextual and personal factors in employee creativity, and based on the understanding, report new areas for research. The research literature in different disciplines on factors that impact on human creativity including creativity in work-related activities is very large, but a systematic review and analysis can be of interest.
One strength of the submission is its systematic approach. It reports a systematic literature review, then uses the established and cited AMO theory to categorise the relevant literatures according to the different factors that can separately and jointly influence human creativity.
Response: Thank you for appreciating the strengths of our work.
However, the submission also has a number of weaknesses that undermine the case for publication in the journal. One is the fit with the journal title – Environmental Research and Public Health – and the theme of Digital Health. Nowhere in the submission is there a clear link to creativity in healthcare, how digital technologies evolve work so that it might be more creative, or creative in different ways, or how digital tools can be used to enhance human creativity. is not cited, and research exploring how digital tools influence human creativity in conferences such as ACM Creativity & Cognition is not cited. Instead, the submission appeared to be written for more general HRM and business studies title. Instead, a more explicit fit of the research to the journal needs to be made – digitization and its impact on workplace creativity is great, and something worth researching, but it needs to recognise and reference relevant previous research.
Response: Thanks for this constructive advice. We have now put much effort into rewriting our rationales and arguments. With respect to justifying our point of creativity in the digital era, we have taken your advice to make the following corrections, additions, improvements:
First, we extended the creativity research towards the journal and issue theme. Specifically, we integrated and cited relevant work on ‘creativity in healthcare’ and relevant research on ‘how digital tools influence human creativity in general’ (e.g., research from the ACM Conference on Creativity and Cognition). For example, at the beginning of the introduction (p. 2, line 44-52), we now refer to Zachos et al. (2015) and Spadotto et al. (2009) to indicate how ICT helps to increase innovation and development of a business operation, as well as that new ICT-applications can and should be creatively used and developed in healthcare. Thus, we followed previous studies, which highlight that ICT can stimulate human creativity. To further strengthen our arguments, we referred to a study from Lazarus (2011) and two studies from Mu et al. (2018, 2019) in our argument that “ICT-supported creativity can support healthcare practice improvement, innovation and operation” (p. 2, line 51-52).
References:
Zachos, K., Maiden, N., & Levis, S. (2015, June). Creativity Support to Improve Health-and-Safety in
Manufacturing Plants: Demonstrating Everyday Creativity. In Proceedings of the 2015 ACM SIGCHI Conference on Creativity and Cognition (pp. 225-234).
Spadotto, E., Hawkins, J., & Monrose, K. (2009, February). ICT convergence, confluence and creativity: The application of emerging technologies for healthcare transformation. In Proceedings of the 3rd international symposium on medical information and communication technology.
Lazarus, I. R. (2011). Innovation or stagnation? Crossing the creativity gap in healthcare. Journal of Healthcare Management, 56(6), 363-367.
Mu, Y., Bossink, B., & Vinig, T. (2018). Employee involvement in ideation and healthcare service innovation quality. The Service Industries Journal, 38(1-2), 67-86.
Mu, Y., Bossink, B., & Vinig, T. (2019). Service innovation quality in healthcare: service innovativeness and organisational renewal as driving forces. Total Quality Management & Business Excellence, 30(11-12), 1219-1234.
Second, we added analysis and discussion topics in the discussion section to reflect on human creativity in the digital era and related to healthcare (p. 14, line 577-588). Specifically, we began the paper with re-highlighting the importance and urgency of understanding creativity in the digital era based on a reflection on the specific studies we now integrated in the paper (e.g. Zachos et al, Spadotto et al., Lazarus et al., and two times Mu et al.).
Third, acknowledging your comment, besides making corrections above throughout the manuscript, we finished our paper with the remark that our take is limited and opens new avenues for research that can be explored in the future (p. 16-17, line 686-698).
Another weakness is the definition of creativity, which fails to build on established refinements of novel and valuable. For example, does the research focus on what Kaufman and Beghetto call Pro-C creativity, and/or does it refer to little-c creativity, or other types? More effective scoping and definition of the central phenomenon under investigation was needed. And given the focus on employee creativity, there was no attempt to differentiate between different types of employee and work role. I might expect to see some differences between roles such as production line assemblage and digital designers or architects. I accept that a detailed analysis of studies by specific roles was not feasible, but a simple separation between roles specific to the creative industries or not, or the level of education needed, might have led to more inciteful findings.
Response: Thank you for pointing out. In the sub-section of 2.1 Definition of Creativity (p. 3-4, line 124-144), we inserted a more fine-grained creativity definition by Theresa Amabile. Specifically, we began with pointing out our research scope in the OB research area as “Grounded in the discipline of organizational psychology, creativity is consistently defined as the employees’ production of novel and useful products in any domain”. Thus, we repeated the definition of creativity from Amabile (1996) to suggest that this definition “emphasis on creativity as an outcome, instead of the mental process through which creative ideas ultimately emerge, allows creativity to be quantified with relative ease and consensus.”
Meanwhile, recognizing that there exists different types of employee and work roles, we further argued that “These scholars propose that creativity is not a uniform construct across all settings; instead, several types of creativity need to be differentiated based on the context in which they were developed. For example, Burns and Stalker were the first to systematically study the differences in working environments that stimulate creative behavior among employees. Through their research they differentiated mechanistic environments that trigger routine thinking and working and so-called organic organizational structures that trigger creative working habits. Next, De Bono was among the first scholars to distinguish creative thinking from routine-based thinking contributing therefore to understanding of the human creative potential in organizations. Based on these research findings, organizational behavior (OB) scholars have been focusing on various leadership explanations triggering employee creativity in organizations. We commend these efforts and argue that they need to be complemented by a more detailed understanding of how different antecedents jointly affect employee creativity. Hence, rather than providing an exhaustive review of the employee creativity literature within a broader range of research background, we aim to take stock of research conducted in line with employee creativity concept as it is used in the OB stream of research.” (p. 3-4, line 130-144).
Finally, we now put forward in the discussion section that our limited scope is a choice and a strength to address your insightful comment. Specifically, we suggested that this opens an avenue for further research that does cover aspects that we left unused by inserting that “With a specific focus on the organizational behavior and HRM research areas, future research ” With a specific focus on the organizational behavior and HRM research areas, future research is encouraged to address the topics of digitization and its impact on workplace creativity (e.g., how digital tools influence human creativity) and creativity in other settings (e.g., creativity in healthcare).” (p. 16, line 686-689).
References:
Amabile, T. M. (1983). The social psychology of creativity: A componential conceptualization. Journal of personality and social psychology, 45(2), 357.
Amabile, T. M., Conti, R., Coon, H., Lazenby, J., & Herron, M. (1996). Assessing the work environment
for creativity. Academy of Management Journal, 39(5), 1154–1184.
Amabile, T. M., Barsade, S. G., Mueller, J. S., & Staw, B. M. (2005). Affect and creativity at work. Administrative science quarterly, 50(3), 367-403.
Bass, B.M. (1990) From transactional to transformational leadership: learning to share the vision. Organizational Dynamics, 18: 19-31.
Burns, T.E., Stalker, G.M. (1961) The Management of Innovation. London: Tavistock.
De Bono, E. (1970) Lateral Thinking: Creativity step by step. New York: Harper & Row.
De Bono, E. (1985) Six Thinking Hats. Boston: Little, Brown & Co.
The submission also treated creativity as unitary concept, which it is not. Research in fields not cited in this paper demonstrates that different basic stages of creative processes (e.g. divergence versus convergence in idea generation) impacted by different individual and contextual factors. The unnecessarily simple definition of creativity adopted for the review limits the rigour of the analysis because it is too coarse-grain. Furthermore, it was not clear to me why team creativity was outside of the scope of the research, given that this how much creativity in the workplace happens.
Response: Thank you for this comment. Consistent with previous point, we cited more creativity papers to indicate the development of creativity literature in terms of definition (see above reaction to your previous comment). We conclude that “We commend these efforts and argue that they need to be complemented by a more detailed understanding of how different antecedents jointly affect employee creativity. Hence, rather than providing an exhaustive review of the employee creativity literature within a broader range of research background, we aim to take stock of research conducted in line with employee creativity concept as it is used in the OB stream of research.” (p. 3-4, line 139-144). Moreover, we added in the text that “We only included studies that concern individual employee creativity rather than team creativity, because the AMO theoretical framework primarily addresses the development of outcomes at the individual employee level (Kim, Pathak, & Werner, 2015)” to highlighted that our research on employee creativity, rather than team creativity, is consistent with the basic assumption of developing employee performance in AMO framework.
In light of these criticisms it was possible to argue that literature review was too narrow and restricted at only 115 from specific journals in the business (but not health or digital disciplines). This amounts to a relatively small number of publications each year since 1990. Of course, a systematic analysis of 115 journal papers is still a substantial effort, and is applauded. However, the number and focus of the papers was only a small fragment of the research reported about human creativity and factors that influence it. Not only are there other sources of research that investigate factors that influence employee creativity, but also there are a larger set of studies that provide evidence towards theories and models of personal and contextual factors that influence human creativity. For example, the 2008 meta-analysis of contextual factors on human creativity by Matthijs Baas and colleagues drew on over 160 papers and revealed patterns about the nature of the space and environment in which people are creating. This clearly has some import to the research? And again, the literature review does not reference digital health or health explicitly.
Response: Thank you for this comment. Given that the current research has a particular question—that is, it specifically targets on the ‘international effects’ of predictors towards employee creativity, we only selected empirical papers which examine the interactions (this can be found in the introduction section and methodology section, p 2, line 93). Thus, unlike previous meta-analytical papers (e.g., Matthijs Baas et al., 2008) which capture insights of general effects of factors (e.g., positive/negative mood) on creativity, we used strict requirements for our search strategy, which only includes papers with an interactional perspective, which provides strong evidence of interactional effects in creativity literature that can inform policy and practice.
We agree that our strategy of selecting the appropriate papers for the targeted research question in our review paper is of a specific focus, and that future research can be encouraged to broaden our research horizon with a more inclusive research question. Thus, we listed it as one of our limitations (p. 16-17, line 694-698) by inserting “our focus on understanding the interactional effects of predictors on employee creativity could have been potentially restrictive toward understanding a full spectrum of interactions in predicting employee creativity. We would like to encourage research to use other theoretical frameworks to explore and explain interactional effects of various factors on employee creativity, thus allowing for further discoveries in this research area.”
The resulting models from the analysis of the literature are relatively simple, and most of the reported analysis returned to instances of cited literature categorised according to the AMO theory. Given my concerns about the source literature used, I struggled to see how the work reported in the paper could provide meaningful guidance to organisations in employee hiring and work redesign.
Response: Thank you for this constructive comment. We understand that you both like our AMO-approach – as stated in the beginning of your review – as well as find it difficult to find the detailed and rigorous meaning and contribution of this approach to practice – as stated in the above, last comment. We understand this, and focused in our revision on further highlighting, explaining, detailing our theoretical approach of applying the AMO framework to review employee creativity literature. Moreover, according to our findings, we highlighted the practical implications of integrating enacting ability-, motivation-, and opportunity-enhancing practices to maximize employee creativity. Specifically, we inserted that” The final implication regards synthetically combining and enacting ability-, motivation-, and opportunity-enhancing practices to enable employee perform creatively. For example, HR should select candidates with digital-orientations, encourage them to utilize technological information resources, and build a supportive work environment to facilitate interactions and communications via digital method among employees.” (p. 16, line 654-658).